# Role of Phase Change Materials Containing Carbonized Rice husks on the Roof-Surface and Indoor Temperatures for Cool Roof System Application

**DOI:** 10.3390/molecules25143280

**Published:** 2020-07-19

**Authors:** Hong Gun Kim, Yong-Sun Kim, Lee Ku Kwac, Mira Park, Hye Kyoung Shin

**Affiliations:** 1Institute of Carbon Technology, Jeonju University, 303 Cheonjam-ro, Wansan-gu, Jeonju-si, Jeollabuk-do 55069, Korea; hgkim@jj.ac.kr (H.G.K.); wva223g6@naver.com (Y.-S.K.); kwac29@jj.ac.kr (L.K.K.); 2Carbon Composite Energy Nanomaterials Research Center, Woosuk University, Wanju, Chonbuk 55338, Korea

**Keywords:** phase change materials, carbonized rice husk, cool roof system, thermal conductivity

## Abstract

This study researches the effect of phase change materials (PCMs) containing carbonized rice husks (CRHs) in wood plastic composites (WPCs) as roof finishing materials on roof-surface and indoor temperatures. A cool roof miniature model was prepared, and measurements were taken using three fixed temperatures of 30 to 32 °C, 35 to 37 °C, and 40 to 42 °C. Sodium sulfate decahydrate (Na_2_SO_4_·10H_2_O) and paraffin wax were selected as the PCMs. CRHs were used as additives to improve the thermal conductivities of the PCMs. At lower fixed temperatures such as 30 to 32 °C and 35 to 37 °C, the rates of increase of the surface temperatures of roofs containing CRHs with Na_2_SO_4_·10H_2_O, and paraffin wax, were observed to gradually decrease compared to those of the roofs without PCMs. The indoor temperatures for the above-mentioned PCMs containing CRHs were maintained to be lower than those of the indoors without PCMs. Additionally, as the CRH content in the PCM increased, the rates of increase of the roof-surface and indoor temperatures decreased due to a faster roof heat absorption by PCMs through the improved thermal conductivity of CRHs. However, under higher artificial temperatures such as 40 to 42 °C, Na_2_SO_4_·10H_2_O with CRHs exhibited no effect due to being out of latent heat range of Na_2_SO_4_·H_2_O. For paraffin wax, as CRH content increased, their roof- surface and indoor temperatures decreased. Especially, the surface temperature of the roof containing paraffin contained 5 wt.% CRHs reduced by 11 °C, and its indoor temperature dropped to 26.4 °C. The thermal conductivity of PCM was enhanced by the addition of CRHs. A suitable PCM selection in each location can result in the reduction of the roof-surface and indoor temperatures.

## 1. Introduction

Direct exposure of construction surfaces to sunlight has been observed to increase the temperature in buildings; this is because of the higher heat storage and lower heat emission of materials such as concrete or metal deck [1,2,3,4,5,6]. This can lead to an urban heat island phenomenon that causes the central city area temperature to be higher than that of suburban areas, thus contributing to global warming [7,8,9,10]. Therefore, employing cool roof systems using phase change materials (PCMs) as building insulation is required. These minimize the heat transfer by limiting it to the interior of building, and can also maintain a stable indoor temperature, while providing the highest energy saving for electricity consumption [11,12,13,14,15] (see in Figure 1).

PCMs used for cool roof system applications are materials capable of absorbing or releasing thermal energy in the form of latent heat during the solid–liquid transition, and are extensively applied in many energy economy areas due to sustained heat source temperature, high energy density, and repeated utilization [16,17,18,19,20]. Therefore, by utilizing the advantages of PCMs, numerous researchers have studied the effect of only using PCMs on roof-surface and indoor-temperature reduction for cool roof systems. Yang et al. [21], Dong et al. [22], and Jayalath et al. [23] researched cool roof systems using PCM without the additives. Regarding the results, PCM roof temperatures were reduced or heat transfer was delayed. Saffari et al. [24] reviewed papers on the numerical simulation of buildings with PCMs for passive cooling using whole building energy simulation tools. Costanzo el al. [25] investigated commercial PCMs employed as mats within drywall partition systems in air-conditioned lightweight office buildings under thermostatic control, and their influence on the indoor operating temperatures and cooling load. 

Nevertheless, PCMs have major disadvantages such as supercooling and phase segregation due to low thermal conductivity, corrosion, and latent heat loss during liquid–solid repetition. Therefore, to inhibit these disadvantages, various additives have to be added to PCMs. Carbonized materials are widely used to improve the thermal conductivity, which results in the prevention of supercooling and phase segregation. Additionally, they possess favorable properties such as noncorrosion by contact with PCMs, chemical stability, nontoxicity, and being lightweight [26,27,28,29,30,31,32,33,34,35,36]. Amongst the carbon materials with these advantages, carbonized rice husks (CRHs) are inexpensive as they are waste materials that can be easily obtained. Additionally, the convex shape of rice husks can maximize the energy storage capability by a simple impregnation method and increase the specific area for heat transfer in PCMs. 

The aim of this study was to research the attainment of reduced roof-surface and indoor temperatures through fast roof heat absorption by improving the thermal conductivity of PCMs through the addition of CRHs. In addition, the study aimed to minimize the energy loss in latent heat storage and retrieval progress by removing the phase segregation that is not completely soluble during melting and the supercooling problems that remain molten PCMs during crystallization to ensure timely release of the heat of fusion. To apply the cool roof system, a cool roof miniature model was prepared. 

## 2. Experimental

### 2.1. Materials

Rice husks were purchased from SAMWHA RICE MILL Co. in Korea. PCMs were used as received; Na_2_SO_4_·10 H_2_O (phase change temperature of 32.4 °C with a purity greater than 98.0%) was received from DAEJUNG Co. (Siheung-si, Gyeonggi-do, Korea), and Paraffin wax (phase change temperature of 48 °C, medical grade) was supplied from WR Medical Electronics Co. (Maplewood, MN, USA). Wood plastic composite (WPC) structure samples were obtained from YES WOOD Co. in Korea.

### 2.2. Preparation of CRHs

The CRHs were carbonized for 1 h at a temperature of 1000 °C in muffle furnace under an atmosphere of N_2_ (99.999%) without a stabilization procedure. The obtained CRHs were used as additives to improve the thermal conductivity of PCMs.

### 2.3. Preparation of PCM/CRH Packs and Set Up of Cool Roof Miniature Model

PCM/CRHs (100 g) packs were prepared by physically mixing the PCM powders evenly with 0, 1, 3, 5 wt.% CRHs, respectively, and then inserting the mixture in polyethylene (PE) bags. Hot pressing the ends of the PE bag enclosed the bags. The end of PE bag containing PCM/CRH were closed using a hot press. Table 1 shows the density of Na_2_SO_4_·10H_2_O/CRHs and paraffin wax/CRHs in PE bags. The density increased with the increase of CRHs. Figure 2a,b exhibits the photographs of raw rice husks and corresponding CRHs that were prepared via carbonization at 1000 °C without a stabilization procedure. The photographs demonstrate that the CRHs experienced a reduction in size due to thermal decomposition while maintaining the convex shape of raw rice husks. The convex shape of CRHs improves their impregnation into the molten PCM, and therefore reduces the latent heat loss by the additives. In addition, the high aspect ratio of the CRHs may also increase the specific area of heat transfer by reducing the distances between CRHs within the PCM. Figure 2c,d show optical microscopy images of the Na_2_SO_4_·10H_2_O/CRHs and paraffin wax/CRHs obtained during cooling from the melt in the PE bags. It can be observed that the CRHs were impregnated within the Na_2_SO_4_·10H_2_O and paraffin wax were uniformly dispersed. Figure 3 depicts the cool roof miniature model. The test measured the upper surface of WPC, as well as the indoor temperatures, using a fixed heat source (at 30 to 32 °C, 35 to 37 °C, and 40 to 42 °C). The fixed heat source was controlled through temperature variation (30 to 32 °C: 725 W, 35 to 37 °C: 855 W, and 40 to 42 °C: 1000 W) by the encoder of a halogen lamp (GEO-MH 1000W (A) B/T, GEO LIGHTING, Anseong, Korea). The temperature variation of the thermocouple was measured through a temperature readout box. The obtained data were automatically recorded and saved on a computer. 

### 2.4. Characterization

X-ray diffraction (XRD) pattern was recorded by a RIGAKU, D/MAX-2500 instrument (Rigaku Corporation, Tokyo, Japan) with CuKα radiation generated at 40 kV and 30 mA at a scan rate of 0.4 °/min. The thermal conductivity was measured using a TPS2500S instrument (Hot Disk, Göteborg, Sweden); the data was acquired using a sensor sandwiched between two identical PCM/CRH composites. The PCM/CRHs composites used for the thermal conductivity measurements were obtained through a melting method. The PCMs were melted at their respective phase change temperatures followed by the addition of 0, 1, 3, 5 wt.% CRHs. The resulting mixtures were placed in a cuboid shape bowls (13 × 13 × 2 cm^3^) and cooled at room temperature to yield PCM/CRH composites that were used to measure the thermal conductivity. The latent heat of the PCM/CRHs was measured using differential scanning calorimetry (DSC 25, TA instruments Inc., DE, USA) at a heating and cooling rate of 1 °C min^−1^ in a nitrogen atmosphere. The samples were prepared by cooling at room temperature after mixing the melted PCM with 0, 1, 3, 5 wt.% CRHs. Approximately 10 mg of the samples was used for the DSC analysis. Estimation was performed based on the respective endothermic areas of the DSC curves during PCM melting with TA Instruments TRIOS v4.4.0 software package. 

## 3. Results and Discussion

### 3.1. XRD and Thermal Conductivity of PCM/CRHs

Figure 4 depicts the XRD patterns of rice husks and CRHs. As shown in Figure 4a, rice husk has characteristic peaks at 2*θ* = 13.5°, 17°, 22°, and 35°, corresponding to the (110), (1$\bar{1}$0), (200), and (040) planes with crystalline regions, such as cellulose, and with a broad amorphous region, such as lignin or hemicelluloses. After carbonization, as depicted in Figure 4b, the CRH diffraction peaks were detected at 2*θ* = 26° and 43°, which are associated with the (002) and (100) planes. The (002) peak results from the graphitization conversion, and thus demonstrated the successful carbonization of the rice husks. Table 2 shows the thermal conductivities of PCM/CRHs. The thermal conductivities of Na_2_SO_4_·10H_2_O and paraffin wax increased with increased CRH content due to the higher contact and more dense per-unit area of CRHs, and the thermal conductivities of pure Na_2_SO_4_·10H_2_O were higher than those of pure paraffin wax. Therefore, thermal conductivities of Na_2_SO_4_·10H_2_O/CRHs are generally higher than those of paraffin wax/CRHs under similar conditions.

### 3.2. Latent heat of PCM/CRHs

Figure 5 depicts the DSC analysis results that quantify the changes in the latent heat value of the PCM according to the CRH content, and its results are summarized in Table 3. The latent heat values of pure Na_2_SO_4_·10H_2_O was 216.13 ± 2.70 kJ kg^−1^ at the melting temperature of T_m_ of 32.4 °C, and paraffin wax was 220.45 ± 1.62 kJ kg^−1^ at T_m_ of 48 °C. Generally, PCMs presented one endothermic peak at melting temperature; however, in the case of paraffin wax, double endothermic peaks were observed. The first peak represents the solid–solid conversion, which indicates the change of the ordered phase into the disordered phase, and the second peak, which is the main peak, indicates the solid–liquid conversion. Therefore, in Figure 4a,b, the latent heat range of paraffin wax (17~56 °C) is wider than that of Na_2_SO_4_·10H_2_O (27~41.5 °C). The latent heat values of PCMs gradually decreased as the CRH content was increased in Figure 4a,b and Table 3. During cooling, the solidifying temperatures of Na_2_SO_4_·10H_2_O and paraffin wax increased with an increase of CRHs. In Figure 5a, the exothermic peak for pure Na_2_SO_4_·10H_2_O was observed at approximately 0 °C; however, Na_2_SO_4_·10H_2_O-containing CRHs crystalized at a higher temperature of approximately 10 °C. This is because the thermal conductivity of Na_2_SO_4_·10H_2_O is more improved with the increase of CRHs. In addition, as shown in Figure 4b, pure paraffin wax and paraffin wax/CRHs crystallized near the T_m_ and at a slightly higher temperature with an increase in CRHs. These results indicate that the addition of CRHs, especially in the case of Na_2_SO_4_·10H_2_O, might prevent the supercooling and phase segregation difficulties by improving the thermal conductivity of PCMs.

### 3.3. Influence of Na_2_SO_4_·10H_2_O/CRH in the WPC on the Roof-Surface and Indoor Temperature of the Cool Roof Miniature Model

Figure 6 illustrates the effect of Na_2_SO_4_·10H_2_O/CRHs using the different fixed temperatures, on the roof-surface and the indoor temperatures. As depicted in Figure 6a,b, the surface temperature of the roofs with no PCM increased more steeply than those of the roofs with only Na_2_SO_4_·10H_2_O or Na_2_SO_4_·10H_2_O/CRHs. These temperatures in Figure 6a,b were maintained to be steady at approximately 59 °C and 70 °C, respectively. The temperatures of the indoor temperature with no PCM increased to 29 °C. Concurrently, the roof temperature of the Na_2_SO_4_·10H_2_O-containing CRHs showed a lower increase than that for roof with only Na_2_SO_4_·10H_2_O. Indeed, as the CRH content increased, the roof-surface temperature increased at a slower pace and resulted in a lower overall temperature, compared to the roof with no PCM, and the indoor temperature showed a decreasing trend. These results are due to a faster roof heat absorption through the improved thermal conductivity of CRHs; this led to a faster melting of Na_2_SO_4_·10H_2_O. However, at the fixed temperature of 40 to 42 °C in Figure 6c, all roof surface temperatures reached approximately 80 °C. Indeed, after 1000 s of artificial temperature measurements at the fixed temperature of 40~42 °C, the surface temperature of the roof with Na_2_SO_4_·H_2_O comprising 5 wt.% CRHs was observed to have only a slight difference of approximately 1°C, compared to the roof without Na_2_SO_4_·10H_2_O. This is due to it being out of range of the Na_2_SO_4_·10H_2_O latent heat (see Figure 5a). Therefore, it is evident that there was no effect on the roof-surface temperature regardless of whether or not Na_2_SO_4_·10H_2_O or Na_2_SO_4_·10H_2_O with CRHs were used at the fixed temperature of 40 to 42 °C; however, the indoor temperatures were observed to keep lower for roofs containing Na_2_SO_4_·10H_2_O or Na_2_SO_4_·10H_2_O with CRHs than those of roof without PCM.

### 3.4. Influence of Parffin Wax/CRH in the WPC on the Roof-Surface and Indoor Temperature of the Cool Roof Miniature Model

Figure 7 illustrates the effects of paraffin wax with CRHs using a fixed heat source (30 to 32 °C, 35 to 37 °C, and 40 to 42 °C) on the temperatures of the roof-surface and the indoor temperature of the roof. The roof-surface and indoor temperatures containing paraffin wax with CRHs were observed to increase less than those of roofs without PCM. Additionally, as the CRH content increased, roof-surface temperatures increased at a slower pace, and the indoor temperatures were maintained to be lower; this was similar to the temperatures of the roof surface and the indoor containing Na_2_SO_4_·10H_2_O with CRHs. In the cases of the fixed temperatures of 30 to 32 °C and 35 to 37 °C, when compared to roofs without PCM, the roof-surface temperatures with paraffin wax with CRHs and without PCMs showed a maximum difference of approximately 7 °C and 11 °C, respectively; further, the indoor temperatures exhibited a maximum difference approximately 4 °C and 6 °C, respectively. However, at the fixed temperature of 40 to 42 °C, as the CRH content increased, the roof-surface temperatures decreased. In particular, the surface temperature of roofs containing paraffin wax with 5 wt.% CRHs reduced to 74.6 °C; further, the indoor temperature also decreased to 26.4 °C, even though the roof-surface and indoor temperatures without PCM reached 86 °C and 35.5 °C, respectively. Therefore, it has been concluded that selection of PCMs with higher T_m_ than the summer temperatures of each location can significantly influence the reduction of roof-surface temperatures and indoor temperature in buildings.

## 4. Conclusions

This study aimed to obtain lower roof-surface and indoor temperatures through the fast heat absorption of PCMs in WPC of roof finishing materials; this was achieved by improving the thermal conductivity of PCMs through CRHs. Under the fixed temperature of 30 to 32 °C and 35 to 37 °C, the surface temperature of the roofs containing Na_2_SO_4_·10H_2_O and paraffin wax with CRHs increased at a slower rate than those of the roofs without PCMs. The indoor temperatures of the roofs containing Na_2_SO_4_·10H_2_O and paraffin wax with CRHs were also maintained to be lower than those of the roofs without PCMs. Additionally, as the CRH content in the PCM increased, the rate of increase of the roof-surface and indoor temperatures also decreased because of a faster heat absorption of the roof surfaces by PCMs due to the improved thermal conductivity through CRHs. However, under a fixed temperature of 40 to 42 °C, the surface temperature of the roof containing Na_2_SO_4_·10H_2_O with 5 wt.% CRHs increased to approximately 80 °C, which was similar to that of the roof without PCM. This was due to being out of range of the latent heat of Na_2_SO_4_·10H_2_O; however, the indoor temperatures containing Na_2_SO_4_·10H_2_O with CRHs were still maintained to be lower than those of the roofs without PCMs. In the case of paraffin wax with Tm of 48 °C and under the fixed temperature of 40 to 42 °C, the CRH content was observed to increase, and the roof-surface temperatures decreased. In particular, when compared to roofs without PCMs, the surface temperature of roof containing paraffin wax with 5 wt.% CRHs reduced to 74.6 °C of a maximum difference of approximately 11 °C and its indoor temperature was also lowered to 26.4 °C. In this study, the improvement of the thermal conductivity of PCM by the addition of the CRHs and a suitable PCM selection in each location can result in the reduction of the roof-surface and indoor temperatures.

## Figures and Tables

**Figure 1 molecules-25-03280-f001:**
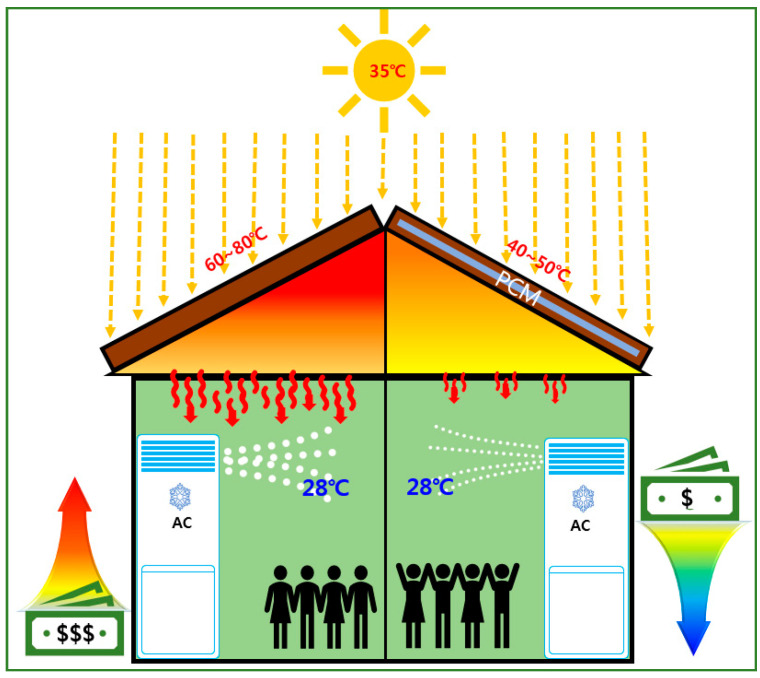
Schematics for the effect of cool roof system using phase change materials (PCM) on energy consumption saving.

**Figure 2 molecules-25-03280-f002:**
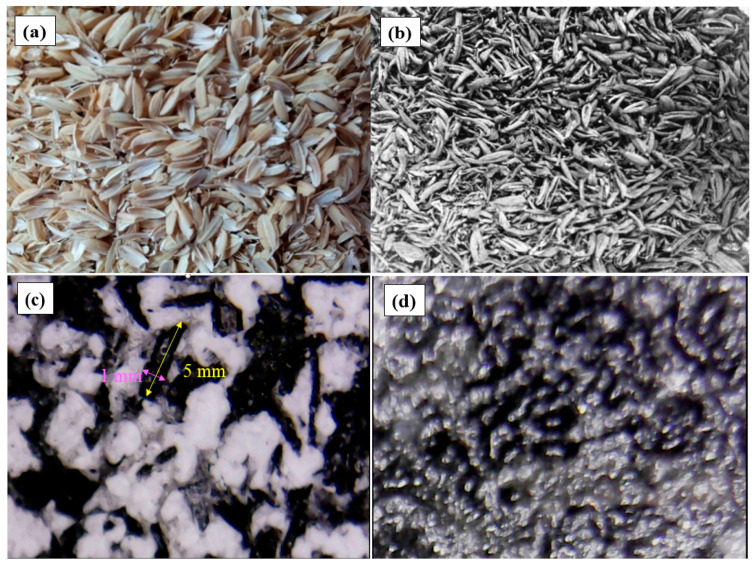
Photographs of (**a**) raw rice husks and (**b**) CRHs. Optical microscopy images of (**c**) Na_2_SO_4_·10H_2_O/CRHs and (**d**) paraffin wax/CRHs (magnification: 200×).

**Figure 3 molecules-25-03280-f003:**
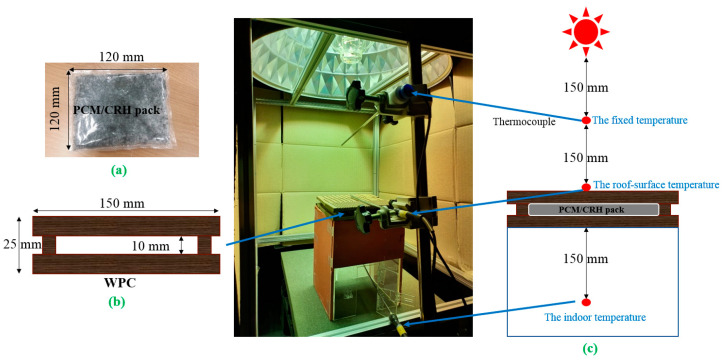
Set up of cool roof miniature model; (**a**) photo of phase change materials (PCM)/CRH pack; (**b**) schematic diagram for cross-section for the wood plastic composite (WPC) cross-section; and (**c**) thermocouple location for temperature measurement.

**Figure 4 molecules-25-03280-f004:**
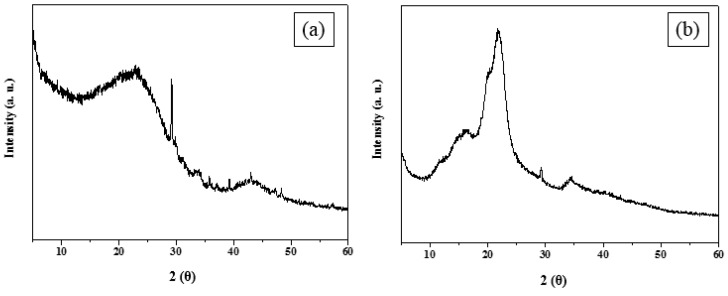
X-ray diffraction (XRD) patterns of (**a**) raw rice husks and (**b**) CRHs.

**Figure 5 molecules-25-03280-f005:**
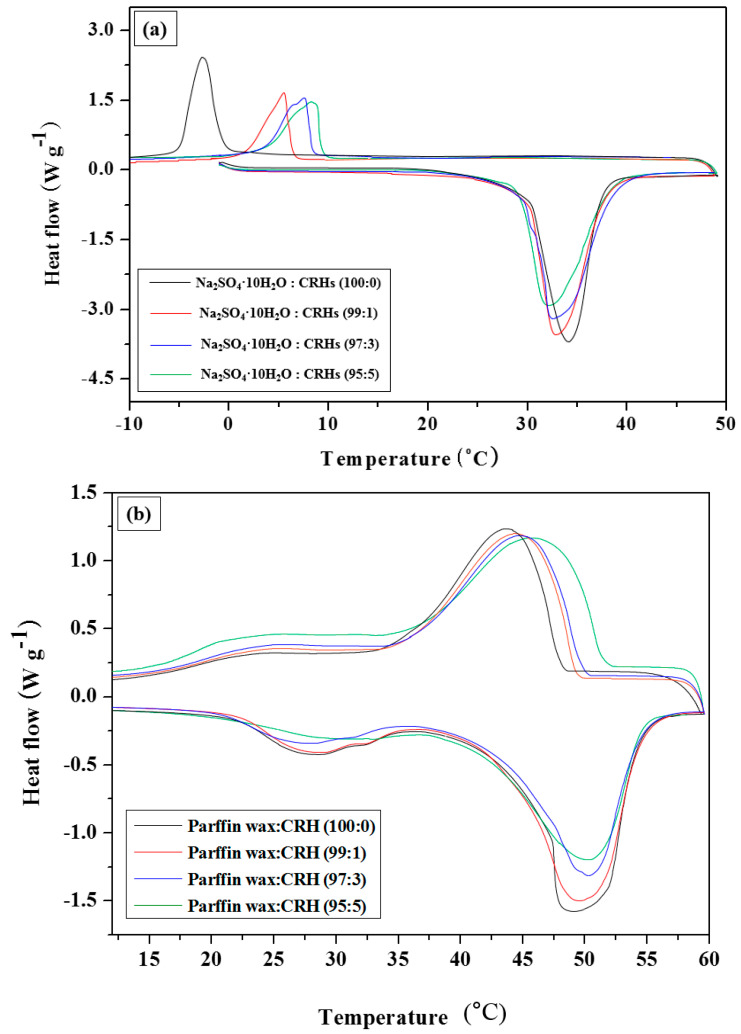
Differential scanning calorimetry (DSC) analysis of (**a**) Na_2_SO_4_·10H_2_O and (**b**) paraffin wax according to wt.% of CRHs.

**Figure 6 molecules-25-03280-f006:**
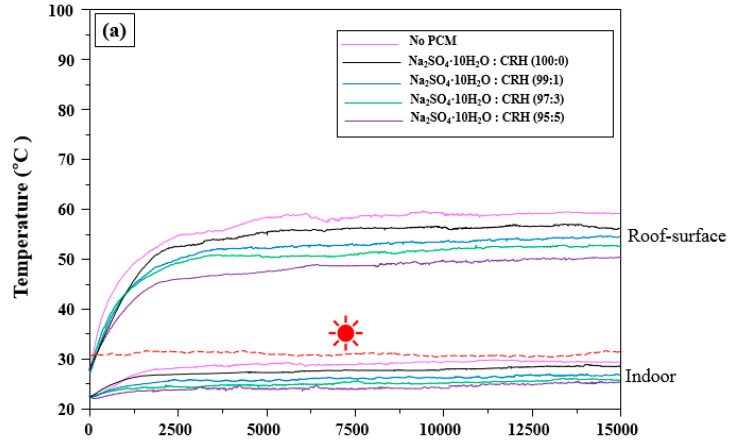
Time-temperature graphs for the temperature at the surface of the roof and the lower part temperatures of a roof containing Na_2_SO_4_·H_2_O/CRHs using a fixed heat source (
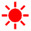
 depicts the fixed temperature): (**a**) 30 to 32 °C, (**b**) 35 to 37 °C, and (**c**) 40 to 42 °C.

**Figure 7 molecules-25-03280-f007:**
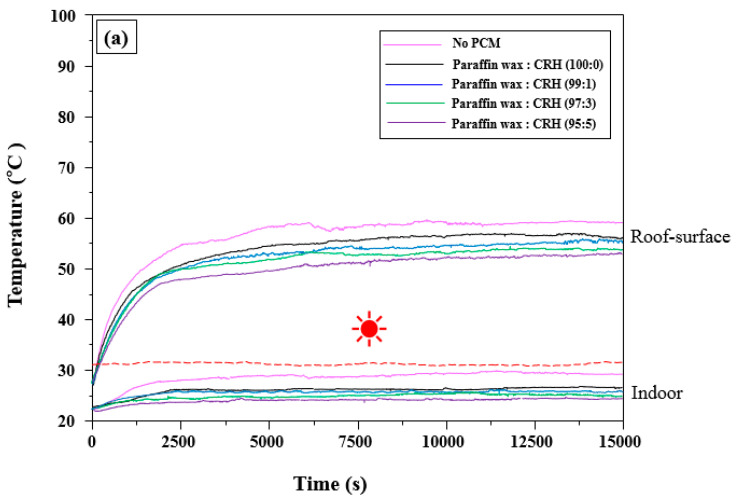
Time–temperature graphs for surface and lower part temperatures of roofs containing paraffin wax/CRHs using a fixed heat source (
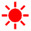
 depicts the fixed temperature: (**a**) 30 to 32 °C, (**b**) 35 to 37 °C, and (**c**) 40 to 42 °C).

**Table 1 molecules-25-03280-t001:** Density of Na_2_SO_4_·10H_2_O/CRHs and paraffin wax/carbonized rice husks (CRHs).

PCM	PCM:CRH wt.%			
Na_2_SO_4_·10H_2_O:CRHs	100:0	99:1	97:3	95:5
Density (kg m^−3^)	858.88 ± 15.6	867.55 ± 3.3	883.00 ± 7.0	894.80 ± 4.2
Paraffin wax:CRHs	100:0	99:1	97:3	95:5
Density (kg m^−3^)	450.12 ± 8.4	477.07 ± 6.2	501.56 ± 6.9	521.25 ± 9.3

**Table 2 molecules-25-03280-t002:** Thermal conductivity of PCM/CRHs.

PCM	PCM:CRH wt.%			
Na_2_SO_4_·10H_2_O:CRHs	100:0	99:1	97:3	95:5
Thermal conductivity (W mK^−1^)	0.718 ± 0.007	0.826 ± 0.008	1.01 ± 0.007	1.43 ± 0.004
Paraffin wax:CRHs	100:0	99:1	97:3	95:5
Thermal conductivity (W mK^−1^)	0.14 ± 0.013	0.22 ± 0.012	0.34 ± 0.009	0.40 ± 0.009

**Table 3 molecules-25-03280-t003:** Latent heat changes of phase change materials (PCMs) according to wt.% of CRHs.

PCM	PCM:CRH wt.%			
Na_2_SO_4_·10H_2_O:CRHs	100:0	99:1	97:3	95:5
Latent heat (kJ kg^−1^)	216.13 ± 2.70	201.62 ± 1.03	192.45 ± 1.78	185.28 ± 2.01
Paraffin wax:CRHs	100:0	99:1	97:3	95:5
Latent heat (kJ kg^−1^)	220.45 ± 1.62	208.23 ± 2.01	194.55 ± 2.45	188.09 ± 1.47

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
