# Peer review of "Role of Phase Change Materials Containing Carbonized Rice husks on the Roof-Surface and Indoor Temperatures for Cool Roof System Application"

_molecules, 2020, doi:10.3390/molecules25143280_

Round 1
Reviewer 1 Report
The manuscript by Hong Gun Kim et al. titled “Role of phase change materials containing carbonized rice husks on the roof-surface and indoor temperatures for cool roof system application” discuses new composite phase change materials for the roof-surface application.
The manuscript is written in a good manner, easy to read and understandable. Research is organised and presented in a concise manner. Presented study has some novelty as addition of carbonized rice husks to the well-known and widely studied PCMs (Na2SO4*10H2O and paraffin wax). The scope of this research fits to the journal guidelines. I would recommend this manuscript for publication after major revision. Few of the points to address:
- Please add the sentence describing the novelty of this work.
- Why authors chose these two PCM for their study? Na2SO4*10H2O shows very high supercooling and with crystallization temperature below 10C it has no practical application. Please explain.
- Please explain the procedure of PCM/CRHs mixing together?
- Please explain the purpose of PE bags?
- Please provide DSC procedure (heating and cooling rate) ideally that should be presented as 1C/min to mimic “real life” conditions where the material would be very slowly heated. If different please provide DSC for 1C/min heating/cooling rate.
- Where the PCMs materials were purchased from? purity?
- Would be valuable to see the roof-surface and indoor temperature test using only pure CHR.
- Time-temperature graphs are shown for the time max 15000s which is just about 4h. Will this material keep the indoor temperature for longer? In the “real life” conditions this material would be exploit to high temperatures much longer than 4h. Would be valuable to present longer test (12h) just for the best composite material. Please add.
Author Response
Thank you for the reviewers for their very constructive comments on this manuscript.
Revision was done as follows.
- Please add the sentence describing the novelty of this work.
- → I revised as your comments
After: The aim of this study was to research the attainment of reduced roof-surface and indoor temperatures through fast roof heat absorption by improving the thermal conductivity of PCMs through the addition of CRHs. In addition, the study aimed to minimize the energy loss in latent heat storage and retrieval progress by removing the phase segregation that is not completely soluble during melting and the supercooling problems that remain molten PCMs during crystallization to ensure timely release of the heat of fusion. To apply the cool roof system, a cool roof miniature model was prepared.
2. Why authors chose these two PCM for their study? Na2SO4*10H2O shows very high supercooling and with crystallization temperature below 10C it has no practical application. Please explain.
→ The reasons that two PCMs were chosen the follows;
- They can make it easily applicable to WPC as roof finishing of existing buildings because they are cheap and easy to obtain.
- Na2SO4∙10H2O has a melting point between approximately 30 °C and 40 °C, which is generally suitable for summer temperature. Paraffin wax has a melting point higher than 40 °C of general summer temperature. Therefore, we chose two PCMs with different melting points to compare the effect of PCM on the roof temperature reduction.
→ In DSC analysis, the solidification temperature was low despite the CRH addition. However, we could be visible that Na2SO4∙10H2O was solidified faster with the increase of CRH. Therefore, according to each location, it can have practical application
3.Please explain the procedure of PCM/CRHs mixing together?→ Therefore, I revised into “PCM/CRHs (100 g) packs were prepared by physically mixing the PCM powders evenly with 0, 1, 3, 5 wt.% CRHs, respectively”.
→ PCM/CRH mixing was prepared by simple mixing method; PCM powders were physically mixed using glass rod according to 0, 1, 3, 5 wt.% CRHs, respectively.
4.Please explain the purpose of PE bags?
→ To prevent the leakage of PCM during melting, PE bags as the container of PCM is very cheap and easy to obtain. In addition, it was easy to insert into WPC as roof finishes if PE bags use.
5. Please provide DSC procedure (heating and cooling rate) ideally that should be presented as 1C/min to mimic “real life” conditions where the material would be very slowly heated. If different please provide DSC for 1C/min heating/cooling rate.
→ As your comments, Figure 5 and Table 3 revised using the results obtained via DSC analysis that performed with 1°C min-1 of heating and cooling rate.
6. Where the PCMs materials were purchased from? purity?
→ I revised into “PCMs were used as received; Na2SO4∙10 H2O (phase change temperature of 32.4 °C with a purity greater than 98.0%) was received from DAEJUNG (Siheung-si, Gyeonggi-do, Korea) and Paraffin wax (phase change temperature of 48 °C, medical grade) was supplied from WR Medical Electronics Co. (Maplewood, MN, USA).”
7. Would be valuable to see the roof-surface and indoor temperature test using only pure CHR.
→In this study, we researched the influence of PCMs that their thermal conductivity enhanced by CRHs on the roof-surface and indoor temperatures for cool roof system application. So, we thought that the roof-surface and indoor temperature test using only pure CHRs didn’t need in this study. However, Time-temperature graph using only CRH didn’t be add it to our paper, but we only added it here for you. Please refer to it.
(figure)
→ The roof surface and indoor temperature with only CRHs increased more than those without PCMs with the increase of the fixed temperature
8. Time-temperature graphs are shown for the time max 15000s which is just about 4h. Will this material keep the indoor temperature for longer? In the “real life” conditions this material would be exploit to high temperatures much longer than 4h. Would be valuable to present longer test (12h) just for the best composite material. Please add.
→ Thank you for your good advice. As your comments, we want to measure time-temperature for long time above 12h but our computer program for time-temperature measurement is recorded up to maximum 4h. In addition, we can’t do anything for 12 h because we have to be observing if the fixed temperature is constantly maintained well. If we have a pilot test chance for cool roof system, as your comments, we will try and want to get better results by long test (12h).
Thank you again for your review of this paper
Best regards.
Dr. Hye Kyoung Shin

Reviewer 2 Report
- Page 1 line 14 Na2SO4∙10H2O. The full chemical should be given before the first presents of the acronyms
- It is recommended to combine some individual sentences in the introduction into paragraph for better readability (Line 44- line 62 in page 2).
- Carbonized materials are widely used as filler for PCMs, more recent literature should be referenced, for example:
- Chen, B.; Han, M.; Zhang, B.; Ouyang, G.; Shafei, B.; Wang, X.; Hu, S. Efficient Solar-to-Thermal Energy Conversion and Storage with High-Thermal-Conductivity and Form-Stabilized Phase Change Composite Based on Wood-Derived Scaffolds. Energies2019, 12, 1283.
- Zhang, B.; Tian, Y.; Jin, X.; Lo, T.Y.; Cui, H. Thermal and Mechanical Properties of Expanded Graphite/Paraffin Gypsum-Based Composite Material Reinforced by Carbon Fiber. Materials2018, 11, 2205.
- Zhang, X.; Wen, R.; Huang, Z.; Tang, C.; Huang, Y.; Liu, Y.; Fang, M.; Wu, X.; Min, X.; Xu, Y. Energy Build. 2017, 149, 463–470.
- Adding fundamental material characterization such as X-ray powder diffraction (XRD) or X-ray photoelectron spectroscopy (XPS) to demonstrate the successful carbonization of rice husks will enrich the manuscript.
- Figure 4 (a), the Na2SO4∙10H2O : CRHs (100:0) sample shows an endothermic peak at around -2 °C, however, the samples with CRHs show a shift of peak position to around 10 °C. Could the author give more discussion about why the peak was shifted?
Author Response
Dear reviewer
Thank you for the reviewers for their very constructive comments on this manuscript.
Revision was done as follows.
Page 1 line 14 Na2SO4∙10H2O. The full chemical should be given before the first presents of the acronyms
→ I revised as your comments;
Before: Na2SO4∙10H2O
After: Sodium sulfate decahydrate (Na2SO4∙10H2O)
It is recommended to combine some individual sentences in the introduction into paragraph for better readability (Line 44- line 62 in page 2).
→ I revised as your comments;
Before: Yang et al. [22] researched cool roof systems using PCM without the additives to reduce the urban heat island phenomenon. Bio25 (phase change temperature: 25 °C) and n-docosane44 (phase change temperature: 44 °C) as PCMs were used and were applied to wood plastic composites (WPC) as roof finishes. As the results, roof surface temperature reduced by the maximum 6.8 °C and lower part temperature of WPC decreased by 7.5 °C.
Dong et al. [23] numerically studied the effect of common roofs and PCM roofs. The results demonstrated that PCM roofs delayed the average temperature peak of the base layer in a room by 3 h.
Jayalath et al. [24] estimated the effect of PCMs in the enhancement of thermal performance and comfort of residential buildings. They found that PCM roofs delayed heat transfer in buildings and provided better thermal comfort in the room.
Saffari et al. [25] reviewed papers on the numerical simulation of buildings with PCMs for passive cooling using whole building energy simulation tools.
Costanzo el al. [26] performed a research of commercial PCMs used as mats within drywall partition systems in air-conditioned lightweight office buildings and their influence on the indoor operative temperature and cooling load under thermostatic control based on dynamic simulations using EnergyPlus. The experiments were carried out in three different locations in Europe with various thickness and melting temperature of PCM mats. As the results, PCMs decreased the peak inside surface temperature by approximately 5 °C and reduced the peak cooling load by 10 ~ 15% according to the PCM thickness and the outdoor climate.
After: Yang et al. [22], Dong et al. [23], and Jayalath et al. [24] researched cool roof systems using PCM without the additives. As the results, PCM roof temperatures were reduced or heat transfer was delayed. Saffari et al. [25] reviewed papers on the numerical simulation of buildings with PCMs for passive cooling using whole building energy simulation tools. Costanzo el al. [26] investigated commercial PCMs employed as mats within drywall partition systems in air-conditioned lightweight office buildings under thermostatic control, and their influence on the indoor operating temperatures and cooling load.
Carbonized materials are widely used as filler for PCMs, more recent literature should be referenced, for example:
1.Chen, B.; Han, M.; Zhang, B.; Ouyang, G.; Shafei, B.; Wang, X.; Hu, S. Efficient Solar-to-Thermal Energy Conversion and Storage with High-Thermal-Conductivity and Form-Stabilized Phase Change Composite Based on Wood-Derived Scaffolds. Energies2019, 12, 1283.
2.Zhang, B.; Tian, Y.; Jin, X.; Lo, T.Y.; Cui, H. Thermal and Mechanical Properties of Expanded Graphite/Paraffin Gypsum-Based Composite Material Reinforced by Carbon Fiber. Materials2018, 11, 2205.
3.Zhang, X.; Wen, R.; Huang, Z.; Tang, C.; Huang, Y.; Liu, Y.; Fang, M.; Wu, X.; Min, X.; Xu, Y. Energy Build. 2017, 149, 463–470.
Adding fundamental material characterization such as X-ray powder diffraction (XRD) or X-ray photoelectron spectroscopy (XPS) to demonstrate the successful carbonization of rice husks will enrich the manuscript.
→ I added references
Figure 4 (a), the Na2SO4∙10H2O : CRHs (100:0) sample shows an endothermic peak at around -2 °C, however, the samples with CRHs show a shift of peak position to around 10 °C. Could the author give more discussion about why the peak was shifted?
→ I revised as your comments;
Before: In Figure 4 (a), the exothermic peak of pure Na2SO4·10H2O was observed below 0 °C; however, Na2SO4·10H2O containing CRHs crystalized at a higher temperature of approximately 10 °C. In the case of paraffin wax, as shown in Figure 4(b), pure paraffin wax and paraffin wax/CRHs crystallized near Tm and at a slightly higher temperature with an increase in CRHs. These results indicated that the addition of CRHs, especially in the case of Na2SO4·10H2O, prevented supercooling and phase segregation due to the improved thermal conductivity of PCMs.
After: In Figure 5(a), the exothermic peak for pure Na2SO4·10H2O was observed at approximately 0 °C; however, Na2SO4·10H2O containing CRHs crystalized at a higher temperature of approximately 10 °C. This is because the thermal conductivity of Na2SO4·10H2O is more improved with the increase of CRHs. In addition, as shown in Figure 4(b), pure paraffin wax and paraffin wax/CRHs crystallized near the Tm and at a slightly higher temperature with an increase in CRHs. These results indicate that the addition of CRHs, especially in the case of Na2SO4·10H2O, might prevent the supercooling and phase segregation difficulties by improving the thermal conductivity of PCMs.
Thank you again for your review of this paper
Best regards.
Dr. Hye Kyoung Shin

Round 2
Reviewer 1 Report
Thank you for addressing all my comments. I support this manuscript for publication